# Roasted *Astragalus membranaceus* Inhibits Cognitive Decline in 5xFAD Mice by Activating the BDNF/CREB Pathway

**DOI:** 10.3390/antiox14101250

**Published:** 2025-10-18

**Authors:** Ji Hye Yoon, Jinyoung Maeng, Yujin Kim, Gidong Koo, Jeong Seok Shim, Sangeun Im, Subin Jung, Jihwan Shin, Sung-Su Kim, Sungho Maeng

**Affiliations:** 1College of East-West Medical Science, Kyung Hee University, Yongin 17104, Republic of Korea; 2Department of Health and Medical Administration. Shingu College, 377 Gwangmyeong-ro, Seongnam 13174, Republic of Korea; 3Department of of Dental Science & Technology, Shingu College, 377 Gwangmyeong-ro, Seongnam 13174, Republic of Korea; 4College of Pharmacy, Dongduk Women’s University, Seoul 02707, Republic of Korea; 5Nature Sense Co., 40D-1009, Imi-ro, Uiwang City 16006, Republic of Korea

**Keywords:** dementia, Alzheimer’s disease, *Astragalus membranaceus*, roasted *Astragali radix*, 5xFAD

## Abstract

Alzheimer’s disease (AD) is a complex pathological process that incurs significant societal costs, yet effective treatments have not yet been developed. Novel compounds targeting β-amyloid, based on the amyloid cascade hypothesis, have failed to demonstrate clinical efficacy. Among natural products with diverse mechanisms, components contained in *Astragali radix* have shown anti-dementia effects in various preclinical studies, including improved cognitive function, reduced β-amyloid levels, and decreased insulin resistance. This study administered a water-extracted roasted *Astragali radix* (RA) to 3-month-old female 5xFAD mice for 3 months, observing changes in cognitive behavior, blood glucose, and neural signaling. RA lowered glucose levels, improved working memory, fear avoidance memory, and spatial memory, and reduced anxiety behavior in 5xFAD mice. In the hippocampus, the protein expression of BDNF and p-CREB/CREB was increased, while p-JNK/JNK was decreased. The effects of RA were similar to unroasted *Astragali radix* in 5xFAD mice, with some components being more abundant. Therefore, RA enhances its taste and aroma, making it suitable for long-term consumption in the form of tea, which could be effective in preventing neurodegenerative diseases such as dementia.

## 1. Introduction

Alzheimer’s disease (AD) is a prevalent neurodegenerative disorder associated with aging and other factors, imposing a significant burden on medical resources and the economy in aging societies [1]. Various hypotheses exist regarding the pathogenesis of AD, including the amyloid cascade hypothesis, tau protein hypothesis, metal ion disorder hypothesis, oxidative stress hypothesis, dysfunctional glucose metabolism, and cholinergic hypothesis [2,3,4]. Based on this theory, many new drugs have been developed to reduce β-amyloid production and aggregation, but no drug has yet been developed that is both sufficiently effective and has minimal side effects [5]. Instead, methods to increase neuronal viability and reduce cell damage have been proposed, and, in particular,, substances from natural products with antioxidant effects that reduce cellular damage and improve neuronal survival such as Astragalus, Ginkgo biloba, ginseng, and Artemisia are attracting attention [6,7].

Empirically used natural products tend to have fewer side effects and are rich in diverse bioactive compounds, enabling multi-component, multi-target effects. These properties provide significant benefits for diseases with complex pathophysiology, such as AD. In traditional oriental medicine, *Astragali radix*, or ‘Huang gi,’ is the dried root of *Astragalus membranaceus* [8]. The Astragalus genus is widely used in folk medicine, dietary supplements, and cosmetics [8]. Over 200 compounds have been identified in Astragalus, known for their antioxidant, anti-inflammatory, anti-aging, cytoprotective, anti-tumor, anti-diabetic, antibiotic, and immune-enhancing properties [9,10]. Astragalus contains 170 saponins, with astragaloside IV known for its antioxidative, anti-aging, anti-inflammatory, anti-asthmatic, anti-diabetic, anti-atherosclerotic activities, and neuroprotective effects [11,12]. About 60 flavonoids, including calycosin, calycosin-7-O-β-D-glucopyranoside (CCGR), and formononetin, are the main active isoflavonoid compounds with anti-inflammatory, antibiotic, antioxidative, anti-cancer, anti-osteoporosis, anti-diabetic, neuroprotective, and cardioprotective effects [9,13]. Approximately 30 species of Astragalus polysaccharides (APSs) have been reported with anti-tumor, anti-aging, anti-diabetic activities, and cardiovascular, neuroprotective, and hepatoprotective effects [14]. Additionally, APS is associated with antimicrobial activity due to immunopotentiation and a probiotic effect on normal intestinal flora [15].

As *Astragali radix* contains numerous beneficial constituents, an increasing number of studies have reported its effects on improving memory and cognitive function [16]. Bioactive compounds in *Astragali radix* have been demonstrated to inhibit β-amyloid production and aggregation, tau hyperphosphorylation, protect neurons against oxidative stress, inhibit neuroinflammation and apoptosis, promote neural stem cell proliferation and differentiation, and reduce mitochondrial dysfunction [17]. However, natural bioactive substances are sensitive to environmental factors, can easily degrade due to light, oxygen, and temperature, or may have low absorption rates. To address these issues, heat processing can increase bioavailability and enhance the pharmacological effects of bioactive components [18]. Roasted *Astragali radix* (RA) deepens its flavor and texture, improves storage properties, and prevents oxidation. Roasting also breaks down cell walls, improving the release of isoflavones and saponins, and, as a result, the amount of saponin and isoflavone increases, showing enhanced antioxidant and anti-inflammatory effects [19].

Based on our preliminary tests comparing the effects of *Astragali radix* and RA, we decided to test the efficacy of water-extracted RA in an AD model using mice. We anticipated that RA would inhibit cognitive symptoms related to β-amyloid accumulation in 5xFAD mice.

## 2. Materials and Methods

### 2.1. Materials

The RA extract was provided by the Department of Herbal Crop Research, National Institute of Horticultural and Herbal Science, and its extraction procedure and composition are described by Ji et al. [19]. Briefly, *Astragali radix* roots were harvested in Jecheon, Chungcheongbuk-do, in 2022. Dried *Astragali radix* root was roasted at 260 °C and extracted twice with 2.4 L of water at 98 ± 2 °C for 7 h. Scopolamine hydrobromide (Cat# S0929) and D-(+)-Glucose (Cat# G7021) were purchased from Sigma-Aldrich (St. Louis, MO, USA). The primary antibodies used were APP (amyloid precursor protein, Abcam, Cat# ab32136, 1:5000), β-amyloid (Cell Signaling, Cat# 8243S, 1:1000), tau (Cell Signaling, Cat# 4019S, 1:2500), phospho-tau (Cell Signaling, Cat# 9632S, 1:5000), BDNF (Cell Signaling, Cat# 60071S, 1:2500), phospho-CREB (Cell Signaling, Cat# 9198S, 1:2500), CREB (Cell Signaling, Cat# 4820S, 1:2000), JNK (c-Jun N-terminal kinase, Abcam, Cat# ab179461, 1:2000), phospho-JNK (Abcam, Cat# ab124956, 1:2500), NeuN (Abcam, Cat# ab104225, 1:2000), synaptophysin (Abcam, Cat# ab14692, 1:5000), GAPDH (Santa Cruz, Cat# sc-32233, 1:5000), and beta-actin (Santa Cruz, Cat# sc-47778, 1:5000). The secondary antibodies used were anti-rabbit IgG (H + L) HRP conjugate (Promega, Madison, WI, USA, Cat# W4011, 1:5000) and anti-mouse IgG (H + L) HRP conjugate (Promega, Madison, WI, USA, Cat# W4021, 1:5000).

### 2.2. Animals and Experimental Groups

To maintain the 5xFAD line, male hemizygous 5xFAD mice aged 8–10 weeks (B6SJL-Tg(APPSwFlLon, PSEN1*M146L*L286V)6799Vas/Mmjax, Strain #034840, Jackson Laboratory, ME, USA) were mated with 8-week-old B6SJL F1 female mice (Jackson Laboratory, Stock No. 100012). The resulting offspring were weaned at 3 weeks of age, and genotyping was performed after the weaning period by polymerase chain reaction using DNA extracted from tail biopsies. The following primers were used: common forward (5′-ACC CCC ATG TCA GAG TTC CT-3′), wild type reverse (5′-TAT ACA ACC TTG GGG GAT GG-3′), and 5xFAD reverse (5′-CGG GCC TCT TCG CTA TTA C-3′).

Female wild-type (WT) and 5xFAD littermates, confirmed by genotyping, were group-housed (n = 6 per cage) and used for experiments at 3 months of age. Experimental groups consisted of wild-type and 5xFAD mice, which were further divided according to RA treatment as follows: WT, WT + RA 500 mg/kg, 5xFAD, and 5xFAD + RA 500 mg/kg (Figure 1).

Animals were maintained under controlled environmental conditions (24 ± 1 °C, 55 ± 5% relative humidity) with a 12 h light/dark cycle (lights on at 6:00 and off at 18:00). Food and water were available ad libitum. All animal studies were conducted in accordance with the NIH Guide for the Care and Use of Laboratory Animals, and the protocols were approved by the Institutional Animal Care and Use Committee of Kyung Hee University (KHGASP-24-155).

### 2.3. Metabolic Changes

Body weight and Food intake: The body weight and food intake of the mice were measured weekly at the same time of day. Food intake was monitored by weighing the chow provided to each cage, and the amount consumed was calculated by subtracting the remaining food from the initial amount. Food efficiency ratio (FER) was calculated as the ratio of body weight gain to food intake over the experimental period.

Blood glucose: Non-fasting blood glucose levels were measured from tail vein blood at 10:00 AM. On the same day, mice were fasted for 6 h, and fasting blood glucose levels were measured again from tail vein blood using the same procedure.

Oral glucose tolerance test (oGTT): To measure glucose tolerance, mice were fasted for 6 h and orally administered a glucose solution at a dose of 1 g/kg. Blood glucose levels were measured from tail vein blood at 15, 30, 60, 90, and 120 min after administration, and these values were used to calculate the area under the curve (AUC).

### 2.4. Behavioral Measures

Mouse behavior was evaluated using the open field test (OFT), Y-maze, elevated Plus maze (EPM), active avoidance (AA), and Morris water maze (MWM) tasks. Behavioral assessments were conducted 8 weeks after the initiation of RA administration, at which time the mice were 5 months old. RA was administered orally 30 min prior to each behavioral test. The tests were performed sequentially in the following order: OFT, Y-maze, AA, and MWM. All behavioral experiments were conducted under controlled illumination (20 lux) and were automatically recorded and analyzed using a video tracking system (SMART v3.0 software and SMART video tracking system, Panlab Harvard Apparatus, Barcelona, Spain).

Open Field Test (OFT): In a white opaque Plexiglas chamber (50 × 50 × 40 cm), mice were gently placed and their movements were filmed and analyzed for an hour. Total distance moved (cm) was compared between experimental groups.

Y maze: The Y-maze test was conducted to assess working memory and cognitive flexibility. The apparatus consisted of three identical arms (35 × 13 × 7 cm) arranged at 120° angles, forming a Y-shape. Visual cues were placed at the end of each arm to facilitate spatial orientation. Each mouse was placed at the end of one arm and allowed to freely explore the maze for 10 min. During the test, spontaneous exploratory behavior was monitored, and the total distance traveled as well as the number of arm entries were recorded to evaluate locomotor activity. Cognitive performance was assessed by measuring the number of spontaneous alternations, defined as successive entries into all three arms without repetition. The following formula was used to calculate of the percentage of spontaneous alternation: [Number of alternation triplets/(Total arm entries−2)] × 100

Elevated Plus Maze (EPM): Apparatus consisted of a cross-shaped maze made with white opaque Plexiglas arms located 40 cm above the ground. Mice were placed in the center of the maze with their heads facing the open arm, and their movement was recorded for 5 min. The time they spent in the open arm was measured.

Active Avoidance (AA): Mice were placed in the active avoidance system chamber (Scitech Korea, Model No. PAAS) and left undisturbed for 2 min. Each session consisted of 20 trials, in which a conditioned stimulus (CS; 2000 Hz tone and light) was presented for 10 s, followed by an unconditioned stimulus (US; 0.3 mA foot shock) with an inter-trial interval of 2 s. Tests were conducted over 4 consecutive days and for each trial, the latency to move to the opposite compartment was measured.

Morris Water Maze (MWM): The test was performed in a round metal tank 180 cm in diameter and 45 cm in height filled with non-transparent water (22 degree Celsius, 30 cm in depth). A platform was submerged 0.5 cm under the water and was hidden to the animals sight by adding white-colored paint into the water. Four symbolic figures were attached to the wall in well-sighted places from the maze. The training lasted 5 days and was performed four times per day. During the 4 daily trials, the starting point differed on each trial. Training sessions continued until the mouse climbed up to find the platform. The time to reach the platform was measured. On the 6th day, the probe test was performed by removing the platform. The swimming trajectories were tracked for a total of 90 s and the time spent in the quadrant where the platform was previously placed was measured.

### 2.5. Western Blotting

Under deep anesthetization by isoflurane, mice were sacrificed and hippocampus tissues were dissected and stored at −80 °C until use. For the analysis of protein expression, tissues were homogenized with lysis buffer containing 1% each of phosphatase inhibitor cocktails 2 and 3 (P5726, P044, Sigma-Aldrich) in protein extraction solution (PRO-PREP, iNtRON Biotechnology, Seongnam, Republic of Korea) and centrifuged at 13,000 rpm for 15 min at 4 °C. Protein concentrations were measured using DC™ Protein Assay Reagents A, B, and S (5000113, 5000114, and 5000115, BIO-RAD, Hercules, CA, USA). After electrophoresis using 10% SDS-polyacrylamide gel, the gel phase proteins were transferred to a PVDF membrane (BIO-RAD, Hercules, CA, USA, Cat# 1620177). Membranes were incubated for 1 h at room temperature in 5% skim milk (BD Difco™, Cat# 212100, Franklin Lakes, NJ, USA) dissolved in TBS-T (0.1% tween 20 in TBS) for blocking. Primary antibodies were diluted in 5% skim milk in TBS-T and incubated overnight at 4 °C. The next day, the membranes were washed thrice for 10 min with TBS-T. Secondary antibodies were diluted with 5% skim milk in TBS-T and incubated for 1 h at room temperature. After washing twice for 10 min with TBS-T, luminescence was induced using a ECL solution (WSE-7120S, ATTO Co., Tokyo, Japan) and the samples were photographed with an EZ-Capture MG (ATTO, Tokyo, Japan). Band density was analyzed using ImageJ 1.51k software (NIH, Bethesda, MD, USA).

### 2.6. Statistical Analysis

All results are expressed as the mean ± S.E.M. Statistical comparisons were performed using ANOVA (blood glucose, OGTT-AUC, OFT, EPM, Y-maze, MWM-probe, density analysis of Western blot images) and repeated measure ANOVA (body weight, OGTT, MWM-latency), followed by LSD or Tukey’s HSD post hoc multiple comparisons testing using SPSS 22 (SPSS Inc., Chicago, IL, USA).

## 3. Results

### 3.1. RA Increased Body Weight and Improved Food Efficiency Ratio of 5xFAD Mice

Three-month-old WT and WT+RA mice gained body weight, but not the 5xFAD mice during the 12-week experimental period. RA increased the weight gain in 5xFAD mice (Figure 2A,B). In 5xFAD mice, the average weekly food intake (Figure 2C) and FER was less than WT mice, but RA improved the FER in 5xFAD mice (Figure 2D).

### 3.2. RA Reduced Fasting Blood Glucose and Improved Glucose Tolerance of 5xFAD Mice

RA decreased the non-fasting and fasting blood glucose concentration in wild type mice and decreased the fasting blood glucose concentration of 5xFAD mice (Figure 3A,B). RA also reduced the peak blood glucose concentration in both wild type and 5xFAD mice, but the AUC was reduced only in 5xFAD mice by RA (Figure 3C,D). There was no significant change in the insulin levels and HOMA-IR among experimental groups (Figure 3E,F).

### 3.3. RA Improved Cognitive Deficits in 5xFAD Mice

Percent alternation in the Y-maze as a measure of working memory was improved by RA in 5xFAD mice (Figure 4A). The latency to escape in the active avoidance paradigm was prolonged in 5xFAD mice and RA treatment normalized the lag of latency in 5xFAD mice (Figure 4B). As a measure of spatial learning and memory, latency to find the platform and the percentage spent in the platform area in the MWM test was also improved by RA in 5xFAD mice (Figure 4C,D). There was no significant difference in locomotor activity in the OFT (Figure 4E). 5xFAD mice showed anxiety-like response in the EPM and RA reduced anxiety-like behavior in both wild type and 5xFAD mice (Figure 4F).

### 3.4. RA Did Not Affect the Expression of Hippocampal β-Amyloid and p-Tau in 5xFAD Mice

The expression of APP and β-amyloid increased in 5xFAD mice, but the expression of Tau and p-Tau remained unchanged in the hippocampus (Figure 5). RA treatment did not affect the amount of APP and β-amyloid accumulated in 5xFAD mice (Figure 5A,B). Also, RA did not affect the amount of Tau or p-Tau (Figure 5C,D).

### 3.5. RA Normalized Altered Neurotrophic Signaling Pathways in 5xFAD Mice

In 5xFAD mice, the hippocampal expression of BDNF and p-CREB/CREB was decreased and p-JNK/JNK was increased (Figure 6). RA increased the expression of BDNF and the p-CREB/CREB and decreased the p-JNK/JNK.

## 4. Discussion

This study shows that RA can improve cognitive symptoms in an AD model mouse by normalizing the expression of BDNF, p-CREB, and p-JNK, and by lowering blood sugar, although it does not reduce the accumulation of β-amyloid.

Recently, many new substances aimed at reducing β-amyloid accumulation in brain tissue to treat AD have been tested in clinical trials, but most have not shown satisfactory effects. For example, Bapineuzumab, an anti-Aβ monoclonal antibody, failed to improve daily life and cognitive functions in patients with mild to moderate AD [20]. Semagacestat, a γ-secretase inhibitor, failed to improve cognitive function and had serious side effects [21]. Vrubecestat, a BACE1 inhibitor, was discontinued in Phase 2/3 clinical trials because it did not sufficiently improve cognitive function [22]. While the amyloid cascade hypothesis is recognized as the cause of AD, these cases suggest that treatments targeting only β-amyloid are insufficiently effective. It is becoming increasingly clear that AD is caused by complex etiologies, and treatments should be developed based on diverse mechanisms [23]. Therefore, natural products with multiple effects represent a useful approach for developing dementia treatments, particularly those targeting neurotrophic factors such as BDNF, which have proven effective in preclinical studies [24].

5xFAD transgenic mice are widely used as a preclinical model of AD, developing neuroinflammation from 2 months of age, with a decrease in neuron numbers from 4 months, and significant reduction in cognitive function by 9 months [25]. In this study, the effects of feeding RA for 3 months to 3-month-old female WT and 5xFAD mice were analyzed both behaviorally and biochemically. 5xFAD mice did not gain weight during this period and showed less average food intake than WT mice. Avoidance memory and spatial memory-related behavior decreased, and anxiety-like behavior increased. The amount of APP, Aβ, and p-JNK expression increased in the hippocampus, while the expression of BDNF and p-CREB decreased. However, Tau and p-Tau expression did not change in the hippocampus.

Previous studies have shown that *Astragali radix* has anti-oxidative, anti-inflammatory, anti-dementia and anti-diabetic effects in various preclinical models [17]. In a study using RA, it activated the AKT/CREB/BDNF pathway in vitro and activated antioxidant enzymes via the Nrf2 pathway, exhibiting neuroprotective effects. In this study, in vivo RA increased body weight, decreased fasting blood glucose, and improved glucose tolerance in 5xFAD mice. Behaviorally, RA improved working memory (Y-maze), avoidance memory (active avoidance), and spatial memory (Morris water maze), and reduced anxiety-like behavior (elevated plus maze). In the hippocampus, RA increased the expression of BDNF and p-CREB and suppressed the expression of p-JNK in 5xFAD mice. However, it had no effect on the expression of APP, β-amyloid, Tau, and p-Tau. Therefore, RA may be related to the effect of protecting cell damage caused by β-amyloid accumulation, such as the neurotrophic effect and protection from oxidative stress [26].

According to the calpain–cathepsin hypothesis formulated by Yamashima (1998) [27], local ischemic damage and oxidative stress in the nervous system induce Ca^2+^ sequestration, which activates calpain, and hydroxynonenal (HNE), a product of lipid periodation, carbonylate Hsp70.1 [28]. Then the activated μ-calpain cleaves carbonylated Hsp70.1, which will destabilize the lysosomal membrane and cause tau aggregation [29]. As a result, discarded proteins within autophagosomes cannot be transported to perikaryal for digestion and accumulates without being degraded by lysosomes [30]. Because the lysosome membrane is unstable, accumulated protein waste within the lysosome bursts into the cytoplasm and outside the cell through the ruptured membrane, becoming the seed of senile plaques [31]. Additionally, cathepsin-D leaks out from lysosomes and damages intracellular organelles [32,33],. In this process, the antioxidant components of RA may reduce the activation of calpain and inhibit lipid peroxidation, thereby preventing a decrease in the amount of Hsp70.

*Astragali radix* contains many ingredients with antioxidant effect, such as flavonoids, saponins, and APS components [11]. Among these components, isoflavonoids such as formononetin, ononin, calycosin-7-β-glucoside, and calycosin were detected in plasma after oral administration in rodents [34,35]. After roasting the *Astragali radix*, the amounts of calycosin, formononetin, and ononin increased, while calycosin 7-glucoside decreased [19]. Calycosin and calycosin-7-glucoside, which have similar known effects, are the components with the highest content in *Astragali radix* and are well absorbed even after oral administration. They reduced β-amyloid, tau, IL-1β, TNF-α, and AChE activities in the hippocampus and increased acetylcholine and glutathione activities in APP/PS1 transgenic mice [36]. Formononetin is involved in APP protein processing and inhibits the production of amyloid proteins, increases amyloid protein clearance through LRP1, and reduces the influx of amyloid proteins into neurons through RAGE inhibition [37]. Ononin is known to have free radical scavenging and HIF-1 regulation effects [17]. *Astragali radix* also contains quercetin, which is known to inhibit β-amyloid production, aggregation, and tau phosphorylation [38]; inhibit AChE activity [39]; and attenuate oxidative stress and neuroinflammation [40]. Despite these previous findings, we did not observe a reduction in β-amyloid after treating with RA 500 mg/kg for 3 months.

Astragaloside IV, a representative saponin component of *Astragali radix*, is a PPARγ agonist that inhibits BACE1, thereby suppressing the production and aggregation of β-amyloid [41]. It is also known to inhibit tau protein hyperphosphorylation [42], have anti-apoptotic [43] and anti-inflammatory effects [44], promote the proliferation and differentiation of neural stem cells [45], and induce BDNF [46]. BDNF was increased by RA in 5xFAD mice, but since there was no reduction in β-amyloid, our results do not align with the known effects of astragaloside IV.

APS is a component that dissolves in water but not well in alcohol, and is found in large quantities in water-extracted *Astragali radix* [47]. APS is known for various bioactivities, including immunomodulatory, antioxidant, antitumor, antidiabetic, antiviral, hepatoprotective, anti-inflammatory, anti-atherosclerotic, hematopoietic, and neuroprotective effects [26]. In APP/PS1 transgenic mice, a dementia model, APS improved cognitive behavior and reduced the effects of metabolic stress but did not reduce β-amyloid accumulation [48]. Instead, it lowered blood sugar and improved insulin resistance [26]. In streptozotocin-induced diabetic rats, APS improved spatial memory, increased hippocampal neurons in the CA1 region, and increased p-CREB, p-NMDA, and p-CaMKII. APS and *Astragali radix* can also inhibit neuroinflammation. Our study using water-extracted RA lowered blood glucose and improved cognitive behavior but did not reduce β-amyloid accumulation, reflecting the known effects of APS.

5xFAD mice exhibit various phenotypes in addition to cognitive dysfunction. According to Forner et al. (2021), 5xFAD male and female mice at 12 and 18 months had lower body weight than wild-type mice, showed less activity in the open field at 18 months, spent more time in the open arm in the EPM at 4, 8, and 12 months, and lasted longer on the rotarod at 4 months [49]. According to Oblak et al. (2021), cholesterol, LDL, and HDL lipoproteins were reduced in 12-month-old male 5xFAD mice compared to wild-type mice, and blood glucose was reduced in 12-month-old female 5xFAD mice [50]. According to our findings, 5xFAD females gained less weight after 3 months of age. At this time, 5xFAD had lower food intake and a lower food efficiency ratio than other groups, indicating low body weight gain compared to food consumption. In response, RA increased body weight, lowered fasting blood glucose, and improved glucose tolerance in 5xFAD during this period. However, there was no change in insulin resistance, as indicated by the HOMA-IR index. In a previous study, *Astragali radix* also improved hyperlipidemia by promoting lipolysis [51]. APS can reduce hyperglycemia, insulin resistance, and inhibit myostatin secretion from skeletal muscle, thereby inhibiting bone muscle loss [52].

RA showed an anti-anxiety effect in the EPM. Regarding the anxiolytic effect, *Astragali radix* reduced anxiety-like behavior and stress hormones in immobilized mice [53,54]. In particular, astragaloside IV improved anxiety and depressive symptoms induced by bisphenol A [55], and showed an anti-depressant effect on chronic unpredictable mild stress [56]. In the OFT, the average locomotor activity of 5xFAD mice was similar to other groups, but variation increased. This may relate to variable activity levels in 5xFAD mice depending on age [25,49]. However, the average activity level did not differ significantly from other groups, indicating that anxiety-like behavior in the elevated plus maze and behavioral changes in latency and active avoidance in the MWM were not confounded by activity level differences, but rather resulted from anxiety and cognitive behavior.

BDNF plays an important role in neuroprotection by reducing β-amyloid toxicity in AD [57]. BDNF expression in the hippocampus was reduced in female 5xFAD mice [58], and increasing BDNF secretion improved cognitive function [59]. Increasing BDNF secretion through astrocytes in 5xFAD mice improved cognitive function not by reducing β-amyloid accumulation or stimulating neuronal regeneration, but by restoring cognitive function through improving the structure and function of the hippocampal neural circuit [60]. Astragaloside IV, formononetin, formononetin-7-O-glucoside, and calycosin interact with TrkB [61]. In a previous study inducing neuroinflammation with lipopolysaccharide, *Astragali radix* reduced microglial activation, BBB dysfunction, and cognitive dysfunction associated with BDNF/TrkB/CREB signaling pathway activation [62]. In our study, BDNF and the downstream effector of BDNF/TrK signaling, p-CREB/CREB, increased with RA in 5xFAD mice but not in WT mice.

JNK plays an important regulatory role in oxidative damage and programmed cell death [63]. The degree of JNK activity correlated with cognitive decline [64], and JNK inhibitors suppressed neuronal cell death, improved cognitive function, and, at high doses, reduced β-amyloid in 5xFAD [65]. Flavonoids from *Astragali radix* inhibited AKT and JNK signaling in LPS-activated BV-2 microglia cells [66]. Astragaloside IV reduced apoptosis by inhibiting the p-JNK/JNK pathway in oxygen glucose deprivation/reoxygenation-induced damage [67]. In vivo, *Astragali radix* inhibited JNK and p-JNK expression in the hippocampus of an LPS-induced depression model [68] and showed neuroprotective effects by inhibiting JNK3 expression in an MCAO model [69]. As shown by Ji et al. (2024), RA was enriched with calycosin, formononetin, and ononin, which may have inhibited p-JNK expression, but other ingredients may also have this effect [19].

The improvement in cognitive impairment by RA may also be related to GRP40 signaling. In diabetic dementia, GRP40 and BDNF expressions in the hippocampus are reduced, and polyunsaturated fatty acids such as docasohexanoic acid activate MAP kinase via GPR40, thereby increasing BDNF expression [70]. The roots and seeds of Astragalus genus are rich in various types of unsaturated fatty acids, which may lead to improvements in cognitive function [71].

Although some studies have shown that *Astragali radix* directly inhibits β-amyloid and p-tau accumulation [17,72], we could not reproduce this effect in our experimental setup. This may be due to insufficient dose or the administration period of RA used. Despite failing to reduce β-amyloid levels, RA may contribute to improving dementia symptoms through reducing hyperglycemia, inducing BDNF, and suppressing the JNK pathway. As RA increased, the concentration of isoflavone components related to antioxidant protein induction increased and ROS production decreased [19]. Especially, caspase-9, caspase-3, and cytochrome C release was inhibited and increased Bcl-2 and decreased Bax expression effectively suppressed oxidative stress induced by Aβ accumulation [19]. These findings suggest that anti-apoptotic and antioxidant effects play a vital role in the cognitive improvement of RA. Furthermore, *Astragali radix* appears non-toxic with long-term administration. Studies in SD rats and Beagle dogs have shown that *Astragali radix* is low in toxicity and does not produce adverse effects even with long-term use [73]. APS was also evaluated as safe for chronic neuropathy due to low toxicity at conventional doses, with no side effects observed even with long-term use [74].

Thera are also many other natural products known to be effective in treating or improving AD symptoms in experimental settings. In addition to RA, Gingko biloba, ginseng, and Artemisia are known for multiple effects such as antioxidant, anti-inflammatory, and neuroprotective effects. Gingko biloba reduced the production of APP and β-amyloid [75], reduced β-amyloid toxicity by inhibiting the formation of β-amyloid fibrils [76], and promoted the degradation of phosphorylated tau protein in lysosomes [77]. Ginseng components suppressed astrocyte activation, increased the expression of BDNF, GDNF, and NGF, and inhibited the activity of acetylcholinesterase (AChE) and BACE1 [78,79]. Artemisia reduced β-amyloid accumulation in the AD mice model, activated ERK/AMPK/GSK/Nrf2 signaling [80], and inhibited BACE-1 and Tau aggregation [81]. However, these natural products are expected to be used as adjunctive therapies rather than as primary treatments for AD.

## 5. Conclusions

This study tested the efficacy of RA in 5xFAD transgenic mice. It improved cognitive function despite not preventing pathological β-amyloid accumulation in these mice. This may be related to the neurotrophic and antioxidant effects protecting cell damage caused by β-amyloid accumulation. Furthermore, RA improved glucose tolerance and BDNF expression, and suppressed the JNK pathway, suggesting its potential use in sporadic AD. RA is a roasted version of *Astragali radix*, enhancing its flavor and aroma. While there were some compositional changes in RA, many effects were similar to the known effects of *Astragali radix*. Furthermore, *Astragali radix* has minimal side effects, suggesting that long-term consumption of RA may help prevent neurodegenerative diseases such as dementia.

## Figures and Tables

**Figure 1 antioxidants-14-01250-f001:**
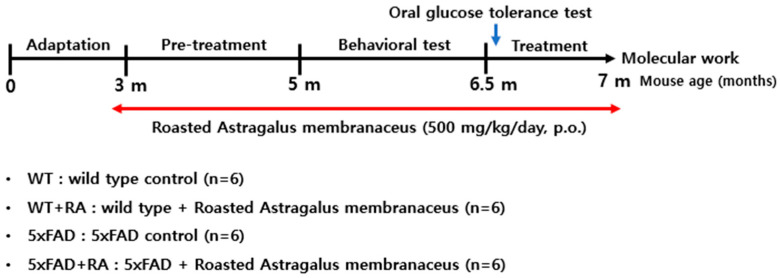
Outline of experimental settings.

**Figure 2 antioxidants-14-01250-f002:**
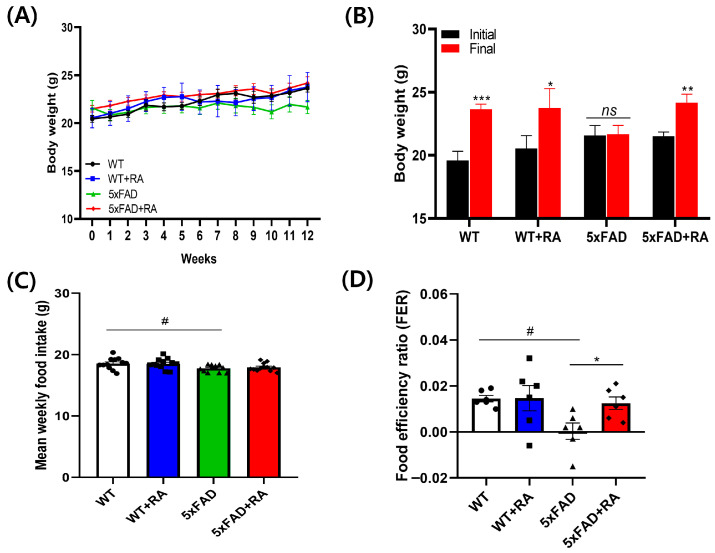
Body weight and food intake during the experimental period. (**A**) Body weight changes over 12 weeks after RA administration (3 to 6 months of age). Repeated measure ANOVA, within group F(2.6, 52.5) = 14, *p* < 0.001, between group F(3, 20) = 0.5, *p* = 0.67, within–between interaction F(7.9, 52.5) = 1.5, *p* = 0.18. (**B**) Body weight change before (week 0) and after (12 weeks) RA administration. 5xFAD+RA initial vs. final; paired *t*-test (t(5) = −4.6, *p* = 0.006. (**C**) Mean food intake during the experimental period. F(3, 44) = 3.3, *p* = 0.03; WT vs. 5xFAD, *p* = 0.015. (**D**) Mean food efficiency ratio (weight gain divided by food intake) during the experimental period. F(3, 20) = 3.6, *p* = 0.03; WT vs. 5xFAD, *p* = 0.012; 5xFAD vs. 5xFAD+RA, *p* = 0.027. All values are mean ± S.E.M. WT: DW fed in wild type females, WT+RA: RA 500 mg/kg fed in wild type females, 5xFAD: DW fed 5xFAD females, 5xFAD + RA: RA 500 mg/kg fed 5xFAD females. * *p* < 0.05, ** *p* < 0.01, *** *p* < 0.001 vs. initial body weight of each group in (**B**), ^#^ *p* < 0.05 vs. WT, * *p* < 0.05 vs. 5xFAD in (**C**,**D**). ns: non-significant. ANOVA, LSD post hoc test.

**Figure 3 antioxidants-14-01250-f003:**
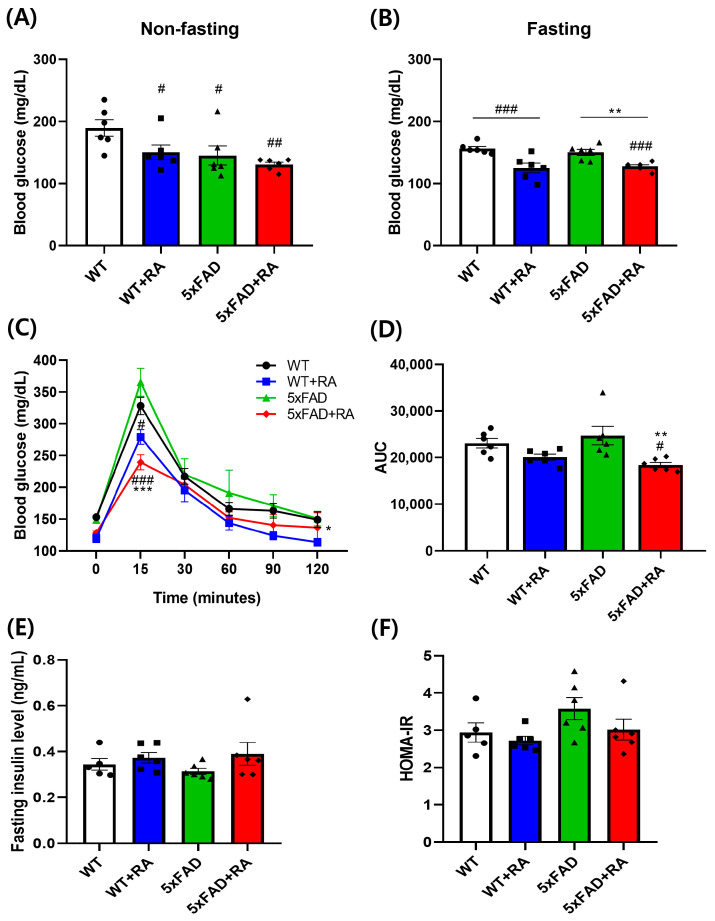
Effects of RA on glucose metabolism on 5xFAD mice. (**A**) Non-fasting blood glucose after 12 weeks of RA treatment. F(3, 20) = 4.5, *p* = 0.015. (**B**) Six hour fasting blood glucose after 12 weeks of RA treatment. F(3, 20) = 9.7, *p* < 0.001. (**C**) Blood glucose concentration changes after 1g/kg of glucose injection in 6 h fasted mice. Within group F(3.2, 64.6) = 198, *p* < 0.001, between group F(3, 20) = 8.3, *p* < 0.001, within–between interaction F(9.7, 64.6) = 2.9, *p* = 0.005. (**D**) Area under curve of oGTT. F(3, 20) = 5.8, *p* = 0.005. (**E**) Fasting insulin level after 12 weeks of RA treatment. (**F**) HOMA-IR after 12 weeks of RA treatment. All values are mean ± S.E.M. WT vs. 5xFAD, *p* = 0.012; 5xFAD vs. 5xFAD+RA, *p* = 0.027. All values are mean ± S.E.M. WT: DW fed in wild type females, WT + RA: RA 500 mg/kg fed in wild type females, 5xFAD: DW fed 5xFAD females, 5xFAD+RA: RA 500 mg/kg fed 5xFAD females. ^#^ *p* < 0.05, ^##^ *p* < 0.01, ^###^ *p* < 0.001 vs. WT, ** *p* < 0.01, *** *p* < 0.001 vs. 5xFAD. ANOVA and repeated measure ANOVA, LSD post hoc test.

**Figure 4 antioxidants-14-01250-f004:**
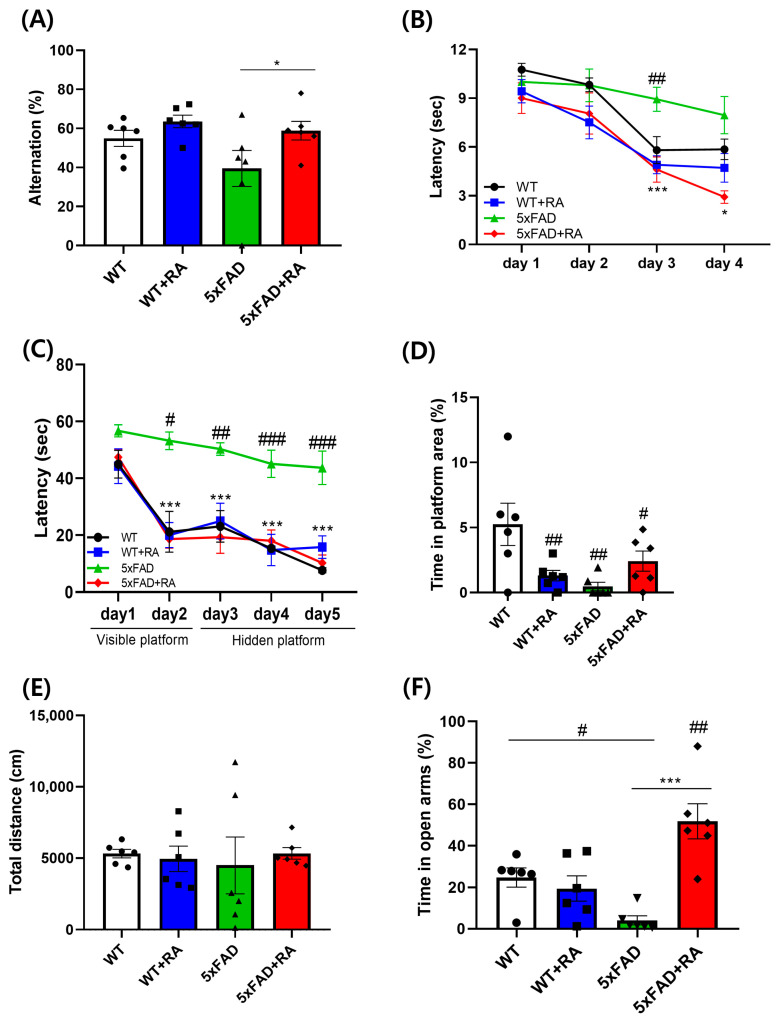
Effects of RA on cognitive behaviors on 5xFAD mice. (**A**) Percent alternation in the Y-maze test. F(3,20) = 3.2, *p* = 0.04. (**B**) Latency to escape in the active avoidance test. Repeated measure ANOVA, within group F(3, 60) = 56, *p* < 0.001; between group F(3, 20) = 4.5, *p* = 0.014; within–between interaction F(9, 60) = 2.9, *p* = 0.007; 5xFAD vs. 5xFAD+RA, *p* = 0.004. (**C**) Latency to find hidden platform in MWM test. Repeated measure ANOVA, within group F(2, 40) = 11.7, *p* < 0.001; between group F(3, 20) = 16.3, *p* < 0.001; within–between interaction F(6, 40) = 1.04, *p* = 0.41; WT vs. 5xFAD, *p* < 0.001; 5xFAD vs. 5xFAD+RA, *p* < 0.001. (**D**) Percentage of time in the platform area in MWM probe trial. F(3, 20) = 4.9, *p* = 0.01. (**E**) Distance moved in the open field test. (**F**) Percentage of time spent in the open arms of the EPM. F(3, 20) = 11.7, *p* < 0.001. All values are mean ± S.E.M. WT: DW fed in wild type females, WT + RA: RA 500 mg/kg fed in wild type females, 5xFAD: DW fed 5xFAD females, 5xFAD+RA: RA 500 mg/kg fed 5xFAD females. ^#^ *p* < 0.05, ^##^ *p* < 0.01, ^###^ *p* < 0.001 vs. WT, * *p* < 0.05, *** *p* < 0.001 vs. 5xFAD. ANOVA and repeated measure ANOVA, LSD post hoc test.

**Figure 5 antioxidants-14-01250-f005:**
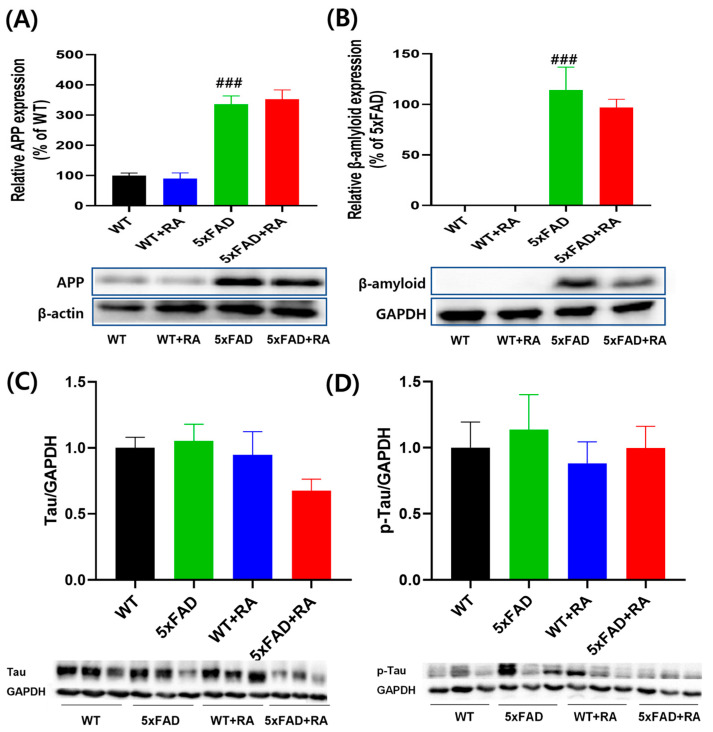
Effect of RA on hippocampal expression on amyloid and Tau proteins in 5xFAD mice. (**A**) Normalized relative expression and representative image of APP. F(3, 16) = 36.2, *p* < 0.001. (**B**) Normalized relative expression and representative image of β-amyloid. No detectable band was observed in WT mice. F(3, 16) = 0.73, *p* = 0.49. (**C**) Normalized relative expression of Tau. F(3, 16) = 1.8, *p* = 0.18. (**D**) Normalized relative expression of p-Tau. F(3, 16) = 0.27, *p* = 0.85. All values are mean ± S.E.M. WT: DW fed in wild type females, WT + RA: RA 500 mg/kg fed in wild type females, 5xFAD: DW fed 5xFAD females, 5xFAD + RA: RA 500 mg/kg fed 5xFAD females. ^###^ *p* < 0.001 vs. WT. ANOVA and LSD post hoc test.

**Figure 6 antioxidants-14-01250-f006:**
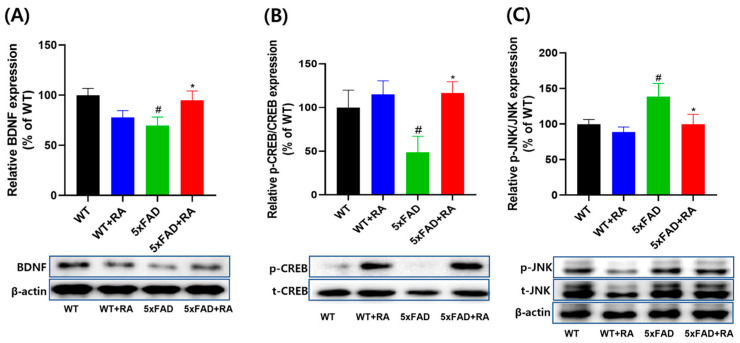
Effect of RA on hippocampal expression of neurotrophic signaling pathways in 5xFAD mice. (**A**) Normalized relative expression and representative image of BDNF. F(3, 20) = 3.3, *p* = 0.04. (**B**) Relative expression and representative image of p-CREB/CREB F(3, 20) = 3.5, *p* = 0.034. (**C**) Relative expression and representative image of p-JNK/JNK. F(3, 20) = 3.1, *p* = 0.049. All values are mean ± S.E.M. WT: DW fed in wild type females, WT + RA: RA 500 mg/kg fed in wild type females, 5xFAD: DW fed 5xFAD females, 5xFAD + RA: RA 500 mg/kg fed 5xFAD females. ^#^ *p* < 0.05 vs. WT, * *p* < 0.05 vs. 5xFAD. ANOVA and repeated measure ANOVA, LSD post hoc test.

## Data Availability

The raw data supporting the conclusions of this article will be made available by the authors on request.

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
