# Peer review of "Roasted Astragalus membranaceus Inhibits Cognitive Decline in 5xFAD Mice by Activating the BDNF/CREB Pathway"

_antioxidants, 2025, doi:10.3390/antiox14101250_

Round 1
Reviewer 1 Report
Using 5xFAD mice model of Alzheimer’s disease, the authors have studied efficacy of roasted Astragalli Radix (AR) for the improvement of cognitive decline. The data are actually interesting but still superficial, and they failed to show and discuss the basic molecular mechanism of this phenomenon.
Intriguingly, the remarkable decline of latency of 5xFAD on days 2-5 was remarkably improved by RA in 5xFAD+RA mice (Fig. 4C). However, the relative expression of APP and beta-amyloid was unchanged between 5xFAD and 5xFAD+RA mice (Fig. 5 A,B). This means that beta-amyloid was not responsible for the cognitive decline in 5xFAD mice. Further, the authors showed normalization of BDNF and pCREB in 5xFAD+RA mice, as compared to 5xFAD mice (Fig. 6).
Citing many literature, they discussed the mechanism of cognitive improvement and normalization of BDNF and pCREB, but not well-done, presumably because they could not cite the appropriate hypothesis of Alzheimer’s disease which stressed the role of reactive oxygen species (ROS) for the development of hippocampal neuronal death. Both CREB phosphorylation and BDNF synthesis were previously reported to be upregulated in response to omega-3 polyunsaturated fatty acids like DHA and EPA via binding with GPR40 in primates. If RA can activate GPR40 (in primates) or fatty acid-bnding proteins (in rodents), one can understand the underlying mechanism precisely.
This paper is still preliminary like a screening test, dealing only with the superficial pharmacological effects of RA, so it needs additional basic analysis. In addition, both anti-oxidative and/or anti-inflammatory effect of RA and its neurotrophic effects had better be discussed, by citing recent hypotheses (for example, calpain-cathepsin hypothesis) about the role of reactive oxygen species in Alzheimer neuronal death. Activated myu-calpain-mediated cleavage of the oxidized Hsp70.1 would cause cathepsin release via lysosomal rupture. It is probable that this calpain-cathepsin cascade may be inhibited by RA.
both anti-oxidative and/or anti-inflammatory effect of RA and its neurotrophic effects had better be discussed, by citing recent hypotheses (for example, calpain-cathepsin hypothesis) about the role of reactive oxygen species in Alzheimer neuronal death. Activated myu-calpain-mediated cleavage of the oxidized Hsp70.1 would cause cathepsin release via lysosomal rupture. It is probable that this calpain-cathepsin cascade may be inhibited by RA.
Author Response
Using 5xFAD mice model of Alzheimer’s disease, the authors have studied efficacy of roasted Astragalli Radix (AR) for the improvement of cognitive decline. The data are actually interesting but still superficial, and they failed to show and discuss the basic molecular mechanism of this phenomenon.
Intriguingly, the remarkable decline of latency of 5xFAD on days 2-5 was remarkably improved by RA in 5xFAD+RA mice (Fig. 4C). However, the relative expression of APP and beta-amyloid was unchanged between 5xFAD and 5xFAD+RA mice (Fig. 5 A,B). This means that beta-amyloid was not responsible for the cognitive decline in 5xFAD mice. Further, the authors showed normalization of BDNF and pCREB in 5xFAD+RA mice, as compared to 5xFAD mice (Fig. 6).
- Yes, we think it's true that APP and beta-amyloid are not the primary cause of cognitive decline., but causing neuronal death which is the cause of cognitive decline. Therefore if a reagent can reduce cell death caused by ABeta or improve the function of neurons, it will be effective in improving cognitive function.
Citing many literature, they discussed the mechanism of cognitive improvement and normalization of BDNF and pCREB, but not well-done, presumably because they could not cite the appropriate hypothesis of Alzheimer’s disease which stressed the role of reactive oxygen species (ROS) for the development of hippocampal neuronal death. Both CREB phosphorylation and BDNF synthesis were previously reported to be upregulated in response to omega-3 polyunsaturated fatty acids like DHA and EPA via binding with GPR40 in primates. If RA can activate GPR40 (in primates) or fatty acid-bnding proteins (in rodents), one can understand the underlying mechanism precisely.
- Although ROS may not be the sole cause of neuronal death, we think the neuroprotective effect of omega-3 polyunsaturated fatty acids via GPR40 is a plausible mechanism for the effect of AR. We discussed the following based on your suggestion:
“The improvement in cognitive impairment by RA may also be related to GRP40 signaling. In diabetic dementia, GRP40 and BDNF expressions in the hippocampus are reduced, and polyunsaturated fatty acids such as docosahexaenoic acid activate MAP kinase via GPR40, thereby increasing BDNF expression [72]. The roots and seeds of Astragalus genus are rich in various types of unsaturated fatty acids, which may lead to improvements in cognitive function [73].”
This paper is still preliminary like a screening test, dealing only with the superficial pharmacological effects of RA, so it needs additional basic analysis. In addition, both anti-oxidative and/or anti-inflammatory effect of RA and its neurotrophic effects had better be discussed, by citing recent hypotheses (for example, calpain-cathepsin hypothesis) about the role of reactive oxygen species in Alzheimer neuronal death. Activated myu-calpain-mediated cleavage of the oxidized Hsp70.1 would cause cathepsin release via lysosomal rupture. It is probable that this calpain-cathepsin cascade may be inhibited by RA.
- As you said, this paper leaves much to be done. In particular, I believe we need to delve deeper into the calpain-cathepsin hypothesis. Our discussion about this is as following :
According to the calpain-cathepsin hypothesis formulated by Yamashima (1998) [28], local ischemic damage and oxidative stress in the nervous system induce Ca2+ sequestration, which activates calpain, and hyrdoxynonenal (HNE), a product of lipid periodation, carbonylate Hsp70.1 [29]. Then the activated m-calpain cleaves carbonylated Hsp70.1, which will destabilize the lysosomal membrane and causing tau aggregation [30]. As a result, discarded proteins within autophagosomes cannot be transported to perikaryal for digestion and accumulates without being degraded by lysosomes [31]. Because the lysosome membrane is unstable, accumulated protein waste within the lysosome bursts into the cytoplasm and outside the cell through the ruptured membrane, becoming the seed of senile plaques [32]. Additionally, cathepsin-D leaks out from lysosomes and damages intracellular organelles [33,34],. In this process, the antioxidant components of RA may reduce the activation of calpain and inhibit lipid peroxidation, thereby preventing a decrease in the amount of Hsp70.
Is the research design appropriate and are the methods adequately described?
Yes
No
Both anti-oxidative and/or anti-inflammatory effect of RA and its neurotrophic effects had better be discussed, by citing recent hypotheses (for example, calpain-cathepsin hypothesis) about the role of reactive oxygen species in Alzheimer neuronal death. Activated myu-calpain-mediated cleavage of the oxidized Hsp70.1 would cause cathepsin release via lysosomal rupture. It is probable that this calpain-cathepsin cascade may be inhibited by RA.
- We agree that the antioxidant action of RA can be explained by calpain-cathepsin hypothesis. In the discussion section, we stated as :
In this process, the antioxidant components of RA may reduce the activation of calpain and inhibit lipid peroxidation, thereby preventing a decrease in the amount of Hsp70.
Reviewer 2 Report
The effect of roasted Astragalus membranaceus roots on the inhibition of cognitive decline in mouse models of Alzheimer’s disease was investigated in this manuscript. The topic fits well to this journal. The study is well-designed and the provided experiments are accurate. The results are promising and meaningful. However, the manuscript also has some drawbacks and thus I recommend major revision.
Introduction: Please mention natural drugs such as Gingko biloba extract used for AD therapy.
Materials: Please also mention here where the AR root was collected/harvested.
Results: A comparison with a positive control drug against AD would be useful to evaluate and underscore the potential of RA as a possible future AD therapy.
Results, Figure 6: The lanes of the blots should be accurately assigned to the corresponding test mouse groups.
Discussion: Please discuss further natural drugs for AD therapy such as Gingko biloba extract. Is RA superior to Gingko biloba?
Discussion: What other natural sources were described for calycosin, formononetin, and ononin which might be considered as therapies for AD?
Author Response
The effect of roasted Astragalus membranaceus roots on the inhibition of cognitive decline in mouse models of Alzheimer’s disease was investigated in this manuscript. The topic fits well to this journal. The study is well-designed and the provided experiments are accurate. The results are promising and meaningful. However, the manuscript also has some drawbacks and thus I recommend major revision.
Introduction: Please mention natural drugs such as Gingko biloba extract used for AD therapy.
- We have mentioned some natural drugs including Gingko biloba as following : “ Instead, methods to increase neuronal viability and reduce cell damage have been proposed, and in particular, substances from natural products with antioxidant effects that reduce cellular damage and improve neuronal survival such as astragalus, ginkgo biloba, ginseng and artemisia are attracting attention [6,7].”
Materials: Please also mention here where the AR root was collected/harvested.
- We mentioned where the AR root was harvested as “AM roots were harvested in Jecheon, Chungcheongbuk-do, in 2022” in the Materials section.
Results: A comparison with a positive control drug against AD would be useful to evaluate and underscore the potential of RA as a possible future AD therapy.
- Yes, a control drug may be useful, but we didn’t consider using a positive control because Alzheimer's disease drugs like donepezil do not prevent amyloid beta accumulation and thus do not control the pathological processes occurring in 5xFAD mice. Therefore, a drug that blocks amyloid beta accumulation should be used as a positive control. However, because the precise mechanism by which amyloid beta is produced is unknown, there wasn’t any suitable positive control drug to use.
Results, Figure 6: The lanes of the blots should be accurately assigned to the corresponding test mouse groups.
- We have added labels to the blot images in figure 5 and 6
Discussion: Please discuss further natural drugs for AD therapy such as Gingko biloba extract. Is RA superior to Gingko biloba?
- In the discussion section : “Many natural products are known to be effective in treating or improving AD symptoms in experimental settings. In addition to RA, Gingko biloba, ginseng, and Artemisia are known for multiple effects such as antioxidant, anti-inflammatory, and neuroprotective effects. Gingko biloba reduced the production of APP and b-amyloid {Yao, 2004 #103}, reduced b-amyloid toxicity by inhibiting the formation of b-amyloid fibrils {Luo, 2002 #104}, and promoted the degradation of phosphorylated tau protein in lysosomes {Qin, 2018 #105}. Ginseng components suppressed astrocyte activation, increased the expression of BDNF, GDNF, and NGF, and inhibited the activity of acetylcholinesterase (AChE) and BACE1 {Lu, 2010 #101}{Choi, 2016 #102}. Artemisia reduced b-amyloid accumulation in AD mice model, activated ERK/AMPK/GSK/Nrf2 signaling (Zhao, 2020 #99), and inhibited BACE-1 and Tau aggregation (Li, 2021 #100). However, these natural products are expected to be used as adjunctive therapies rather than as primary treatments for AD.”
Discussion: What other natural sources were described for calycosin, formononetin, and ononin which might be considered as therapies for AD?
- We described about other ingredients. In AR and RA, Astragaloside IV is a representative saponin component known as a PPARγ agonist that inhibits BACE1 and tau phosphorylation and have anti-apoptotic, anti-inflammatory effects. Also, Astragalus polysaccharides (APS) is known for antioxidant, antidiabetic, anti-inflammatory, anti-atherosclerotic, and neuroprotective effects
Round 2
Reviewer 1 Report
I think, the concept of this paper became much clear by adding two players such as calpain and Hsp70.1 as target of anti-oxidation by Astragali Radix. In addition, by citing the appropriate papers of GPR40 signaling, the authors have well discussed the mechanism of CREB phosphorylation and BDNF synthesis. I think, the basic results and concepts of this paper was strengthened enough to facilitate the following works.
The abbreviations of 'RA' and 'AR' are troublesome and missunderstanding for readers to understand in precise. 'AR' is often utilized in Abstract and Introduction, whereas 'RA' is utilized in Materials & Methods, Results, and Discussion. If 'RA' and 'AR' are distinct, please replace to other proper abbreviation. If the same, please utilize the single one.
Author Response
Thank you for your advice on improving the quality our manuscript. As you suggested, we used the full name as Astragali radix instead of AR, and used the abbreviation RA for roasted Astragali radix.
Reviewer 2 Report
The revised manuscript is suitable for publication now.
n.a.
Author Response
Thank you for your advice on improving the quality of this manuscript.